# Effect of exclusive breastfeeding cessation time on childhood morbidity and adverse nutritional outcomes in Ethiopia: Analysis of the demographic and health surveys

**Dabere Nigatu**[1☯]*, **Muluken Azage**[2☯], **Achenef Motbainor**[2]

1 Department of Reproductive Health and Population Studies, School of Public Health, College of Medicine and Health Sciences, Bahir Dar University, Bahir Dar, Ethiopia, 2 Department of Environmental Health, School of Public Health, College of Medicine and Health Sciences, Bahir Dar University, Bahir Dar, Ethiopia

☯ These authors contributed equally to this work.
* daberen@yahoo.com

**Data Availability Statement:** The authors have no mandate to share the data set or make it publicly available because the data were obtained from The DHS Program repository. However, upon fulfilling

## Abstract

### Background

Though exclusive breastfeeding (EBF) for the first six months is recommended, it remains a debatable issue in both developed and developing countries. Thus, this study investigated the effect of EBF cessation time on childhood morbidity and adverse nutritional outcome in Ethiopia.

### Methods

We used the 2011 and 2016 Ethiopian Demographic and Health Surveys. The study involved 2,433 children under six months of age. Logistic regression model was applied to determine the effect of EBF cessation time on outcome variables. Population Attributable Fraction was calculated to evaluate the public health impacts of EBF termination in the population.

### Results

Discontinuing EBF at 0–3 months (adjusted odds ratio (AOR): 1.95, 95% confidence interval (CI): 1.08, 3.53)) and 4–6 months (AOR: 3.57, 95% CI: 2.19, 5.83) increased diarrhea occurrence compared to children who continued EBF up to 6 months. Children who had terminated EBF at 4–6 months had increased odds of fever (AOR: 1.73, 95% CI: 1.11, 2.68) and acute respiratory illnesses (ARIs) (AOR: 2.74, 95% CI: 1.61, 4.65). Cessation of EBF earlier than 4 months or between 4–6 months was associated with increased odds of having at least one childhood morbidity. Termination of EBF at 0–3 months and at 4–6 months were associated with increased occurrence of wasting (AOR: 2.32, 95% CI: 1.45, 3.74) and underweight (AOR: 2.30, 95% CI: 1.36, 3.91), respectively. Exclusive breastfeeding can avert 42% of diarrhea, 27% of ARI, 21% of fever, 26% of wasting and 23% of underweight burden among children under six months of age.

their criteria, the data can be accessed by requesting from the DHS Program website (Available at: https://www.dhsprogram.com/data/available-datasets.cfm). Upon clicking the webpage, a list of countries name and lists of standard DHS data by year with other details columns would be displayed. When clicking on "Data Available" under the column heading "survey data-sets" with respective survey years (Ethiopia 2011 and Ethiopia 2016), zipped file names under different main recode file names would be displayed. Go to the main recode file name "Children's Recode" check the box to download "ETKR61DT.ZIP" and "ETKR70DT.ZIP" zipped files. The file names require unzipping to run the analysis. The file names used for this analysis are "ETKR61FL.DTA" and "ETKR70FL.DTA". The authors had no special access or privileges to the data source used for this analysis. The data-sets are available to all interested researchers upon request by fulfilling the DHS Program criteria to grant access. Their criteria to grant access to the data-sets are to have a project/research idea with a brief statement of the purpose/aim/objectives.

**Funding:** The author(s) received no specific funding for this work.

**Competing interests:** The authors have declared that no competing interests exist.

## Conclusions

Termination of EBF before six months was associated with increased occurrence of diarrhea, fever and ARIs. It was also linked with increased occurrence of childhood wasting and underweight. The finding emphasized EBF for the first six months to reduce childhood morbidity and adverse nutritional outcomes.

## Introduction

Adequate infanthood nutrition is essential to ensure full potential for growth, development and health of children [1]. In the infant and young child feeding practice, breastfeeding is well recognized intervention since breast milk is uniquely suited to the infant's nutritional needs. It has also an immunological and anti-inflammatory properties that protect against a host of diseases for both mothers and children [2–4]. Moreover, breastfeeding is an effective child health intervention not demanding an extensive health system infrastructure. Hence, an increase in the rates of exclusive and continued breastfeeding can reduce childhood morbidity and mortality inequalities in developing countries [5].

In World Health Organization (WHO) policy documents, exclusive breastfeeding (EBF) for the first six months of age is an articulated public health recommendation to achieve optimal growth, development and health of infants. Thereafter, introduction of nutritionally rich, safe and appropriate complementary feeding and continue breastfeeding up to two years or beyond [6, 7]. Besides, Fewtrell and colleagues remarked the importance of tracking the consequences of the 2001 WHO infant feeding recommendation in different settings to identify and act on adverse events timely [8].

Since the launch of the 2001 WHO policy on exclusive breastfeeding, very widely varying level of compliance and limited progress was observed between and within regions/countries. In 2010, exclusive breastfeeding practice ranged from 3.5% in Djibouti to 77.3% in Rwanda [5] with only 37% proportion of EBF in low-income and middle-income countries [9]. Still studies were inconclusive of the weanling's dilemma: the choice between the known protective effect of exclusive breastfeeding against diseases and the theoretical insufficiency of breast milk alone to satisfy the infant's nutritional requirement beyond four months of age. A cohort study continued supporting to promote EBF for either 4–6 or 6 months [10] while other studies fully accord with the 2001 WHO recommendation of exclusive breastfeeding for the first six months [11–14].

Overall, the literature suggests the importance of research dealing with the effect of EBF for six months on child health and nutritional outcomes to track adverse events of the 2001 WHO breastfeeding recommendation in different settings. The existing weanling's dilemma on the optimal duration of breastfeeding in both developing and developed countries is also appealing for research. Moreover, to the best of the researchers' knowledge, such studies are lacking in Ethiopia. This study, therefore, determined the effect of exclusive breastfeeding cessation time on childhood morbidity and nutritional outcomes in Ethiopia. We also calculated population attributable fraction to show the percentage of adverse child health and nutritional outcomes that could be prevented by making exclusive breastfeeding universal among children under six months of age.

## Methods

### Study design and settings

This study is based on data from the 2011 and 2016 Ethiopian Demographic and Health Surveys (EDHS). The EDHS data is both a nationally and sub-nationally representative survey

based data. The demographic and health survey (DHS) used a questionnaire that designed to collect information from all 15–49 years old eligible women who were residing in the selected households regarding to exclusive breastfeeding.

The EDHS used multi-stage cluster sampling technique to select and include the study participants. The sampling frame used for both 2011 and 2016 EDHS was the census frame created for the 2007 Ethiopia Population and Housing Census (PHC). The census frame had a total of 84,915 complete list of enumeration areas (EAs). For the surveys, the enumeration areas were taken as clusters. An EA is a geographic area covering on average 181 households. The sampling frame contains information about the EA location, type of residence (urban or rural), and estimated number of residential households. Each regions of Ethiopia was stratified into urban and rural areas. In the first stage, urban and rural clusters were selected with probability proportional to EA size from each sampling stratum. Then household listing was done for selected EAs. In the second stage, a fixed number of households per cluster were selected with systematic random sampling technique from the newly created household listing. All women age 15–49 who were either permanent residents of the selected households or visitors who stayed in the household the night before the survey were eligible to be interviewed. Detail information about sample size determination and sampling procedure available in the country DHS reports [15, 16].

In this study, children under six months of age living with their mothers were included while those children under six months of age but not living with their mothers were excluded. A total of 2,433 children under six months of age were involved in the study.

We followed the "Strengthening the Reporting of Observational Studies in Epidemiology (STROBE)" statement in writing the manuscript (S1 Table).

## Outcome variables measurement

The outcome variables include morbidity status and nutritional status. Morbidity status of children were measured based on three morbidities: diarrhea, fever and acute respiratory illnesses (ARIs). In the DHS survey, diarrhea and fever were assessed by asking the following questions: a) has the child had diarrhea in the last 2 weeks? and b) has the child been ill with a fever at any time in the last 2 weeks? Acute respiratory illnesses were assessed based on the women's responses for the following questions: a) has the child had an illness with a cough at any time in the last 2 weeks? b) when the child had an illness with a cough, did he/she breathe faster than usual with short, rapid breaths or have difficulty of breathing? and c) was the fast or difficult breathing due to a problem in the chest or to a blocked or runny nose? The women's response to the above questions were recoded to 'yes' and 'no' options. The presence of ARI symptoms among children was ascertained if a child had a cough accompanied by short, rapid breathing which was chest-related and/or by difficult breathing which was chest-related in the last 2 weeks preceding the survey. Children with at-least one of the three morbidities was determined.

The nutritional status of children was measured by anthropometric z-scores that was assigned based on the WHO Child Growth Standards [17]. Adverse nutritional outcome indicators that this study emphasized were stunting (height for age z-score), underweight (weight for age z-score) and wasting (weight for height z-score). A child was considered as stunted if his/her height for age z (HAZ) score is less than -2.0 standard deviation from the median HAZ of the WHO Child Growth Standards. A child was considered as wasted if his/her weight for height z (WHZ) score is less than -2.0 standard deviation from the median of WHZ of the WHO Child Growth Standards. Underweight, if his/her weight for age z (WAZ) score is less than -2.0 standard deviation from the median on the WHO Child Growth Standards [18–20]. We had also determined a child with at-least one adverse nutritional outcomes.

## Exposure variables measurement

The main exposure variable was exclusive breastfeeding. In DHS, exclusive breastfeeding was assessed by the 24 hours child feeding practice. Children were considered as exclusively breastfed, if their mothers gave them breast milk but nothing else in the 24 hours preceding the interview. Exclusive breastfeeding cessation time was generated for those children who were on exclusive breastfeeding but discontinued it before six months of age. The exclusive breastfeeding discontinuation time was categorized as 0–3 months and 4–5 months.

Variables controlled as confounding variables include maternal age at delivery, place of residence, number of antenatal visits, place of delivery, wealth index quintile, maternal education, sex of child, type of cooking fuel, sanitation facility, source of drinking water, and disposal of child's stools when not using toilet (see the details in S2 Table). The EDHS data set had a variable "wealth index" classified in quintiles after generating wealth index score using principal component analysis. The wealth index score was created based on the number and kinds of consumer goods the household owns, ranging from a television to a bicycle or car; housing characteristics such as source of drinking water, toilet facilities; and flooring materials [15, 16].

## Data analysis

Data were analyzed using STATA version 14.0 (Stata corporation, College Station, Texas, USA) statistical software package. The DHS survey involve a complex survey sampling procedure. Thus, we used sampling design for complex survey sampling analysis.

We used binary logistic regression analysis to control for possible confounding variables. The logistic regression analysis was applied to examine the effect of exclusive breastfeeding cessation time on childhood morbidity and adverse nutritional outcomes. All statistical significances were declared at P-value less than 0.05. We calculated population attributable fraction (PAF) to estimate the contribution of EBF towards reduction of adverse child health and nutritional outcomes. This study is based on a cross-sectional design which is liable for potential confounding factors. Thus, we calculated the PAF using adjusted odds ratio to control potential confounders. The PAF is the proportion of the outcome occurring in the total population of exposed and unexposed individuals attributable to the given exposure. Preventable fraction is PAF for preventive exposures, which imply the fraction of all cases that would be prevented if the whole population were exposed. It was calculated by the formula $PAF = \frac{Pe(OR-1)}{1+Pe(OR-1)} =$ proportion of cases exposed $* \frac{OR-1}{OR}$ and detail expressed somewhere else [21–23]: Where; $P_e$ is exposure prevalence, OR is for continued exclusive breastfeeding up-to 6 months compared with who terminated EBF before 6 months.

## Ethical considerations

We used secondary data. Both the 2011 and 2016 Ethiopian DHS data set were accessed after getting permission from The DHS Program. These data were collected in line with national and international ethical guidelines. Interested reader can refer the 2011 and 2016 EDHS country reports for further reading on the survey protocol.

## Results

### Socio-demographic characteristics

A total of 2,433 (1,248 from the 2011 EDHS and 1,185 from the 2016 EDHS) children under six months of age were involved in the analysis. Three fourth of the mothers' age at delivery were in the age range of 19–34 years with mean maternal age of 27.3 (95% CI: 26.9, 27.7) years.

About 62% of the mothers had no education, and 88% of them were rural residents. Forty five percent of the mothers had no antenatal care follow-up. Seventy six percent of the deliveries were occurred at home. Fifty one percent of the babies were male and 87% of the babies were not weighted at birth (Table 1).

### Breastfeeding, morbidity and nutritional status of children

Fifty five percent of children under six months of age were exclusively breastfed. From those babies who started exclusive breastfeeding, 21.6% terminated EBF in 0–3 months and 21.9% terminated in 4–6 months of age. Children's morbidity status: 9.0% had diarrhea, 14.8% had fever and 6.3% had symptoms of acute respiratory illnesses. Overall, 20.6% of the babies had at least one of the three adverse health outcomes (Table 2).

Regarding to the nutritional status of children 12.9% of the babies were stunted, 14.2% were wasting and 10.8% were underweight. Twenty eight percent of the children had at least one adverse nutritional outcomes (Table 2).

### Effect of exclusive breastfeeding on adverse child health and nutritional outcomes

This study indicated that the odds of children who had terminated exclusively breastfeeding in the age between 0 and 3 months to have diarrhea was 1.95 times higher than exclusively breastfed children (AOR = 1.95, 95% CI: 1.08,3.53). The odds of children who terminated exclusively breastfeeding in the age between 4 and 6 months to have diarrhea was 3.57 times higher than exclusively breastfed children (AOR = 3.57, 95% CI: 2.19,5.83). In the same way the odds of children who had terminated exclusively breastfeeding in the age between 4 and 6 months to have fever was 1.73 times higher than children who exclusively breastfed (AOR = 1.73, 95% CI: 1.11, 2.68). An association also found between termination of EBF and acute respiratory illness symptoms. The odds of children who had terminated EBF in the age between 4 and 6 months to get ARI was 2.74 times higher than children who exclusively breastfed (AOR = 2.74, 95% CI: 1.61,4.65). Cessation of exclusive breastfeeding earlier than 4 months (AOR = 1.66, 95% CI: 1.13,2.43) or between 4–6 months (AOR = 2.15, 95% CI: 1.48,3.11) were associated with increased odds of having at least one of the three adverse child health outcomes (Table 3).

Exclusive breastfeeding can prevent 42% of diarrhea, 21% of fever and 27% of ARI burden among children if we made it universal among under six months of age babies. If we ensure exclusive breastfeeding for the first six months, we can reduce 26% of the burden from the three childhood morbidities (Table 3).

Termination of EBF earlier than 6 months were associated with adverse nutritional outcomes. The odds of children who terminated EBF 0–3 months of age who have wasting was 2.32 times higher than children who did not terminated up to 6 months of age (AOR = 2.32, 95% CI: 1.45, 3.74). We also identified that cessation of exclusive breastfeeding in the age between 4 and 6 months were more likely to result in underweight (AOR = 2.30, 95% CI: 1.36,3.91). Childhood stunting and termination of EBF earlier than 6 months of age didn't show significant association in the bi-variable logistic regression model. There was also no association between termination of EBF earlier than 6 months and having at least one of the three adverse nutritional outcomes. The study demonstrated that ensuring EBF for the first six months of age can avert 26% of childhood wasting and 23% of underweight (Table 4).

### Discussion

The finding of this study demonstrated that early termination of exclusive breastfeeding had effect on childhood morbidity and adverse nutritional outcomes. Termination of exclusively

**Table 1. Sociodemographic characteristics of mothers and children and household facilities, 2011 and 2016 EDHS, Ethiopia.**

| Variables | Frequency (%) |
|---|---|
| Maternal age at delivery | |
| < = 18 | 180 (7.4) |
| 19–34 | 1,824 (75.0) |
| > = 35 | 429 (17.6) |
| Maternal education | |
| No education | 1,507 (61.9) |
| Primary | 743 (30.5) |
| Secondary | 145 (6.0) |
| Higher | 38 (1.6) |
| Place of residence | |
| Urban | 291 (12.0) |
| Rural | 2,142 (88.0) |
| Antenatal care visit(n = 2,430) | |
| No visit | 1,092 (44.9) |
| 1–4 visit | 1,033 (42.5) |
| >4 visits | 305 (12.6) |
| Place of delivery | |
| Home | 1,859 (76.4) |
| Health facility | 574 (23.6) |
| Household wealth status | |
| poorest | 575 (23.6) |
| poorer | 555 (22.8) |
| middle | 501 (20.6) |
| richer | 430 (17.7) |
| richest | 372 (15.3) |
| Sex of child | |
| Male | 1,249 (51.3) |
| Female | 1,184 (48.7) |
| Type of cooking fuel (n = 2,341) | |
| Clean fuel | 45 (1.9) |
| Solid fuel | 2,296 (98.1) |
| Sanitation facility (n = 2,341) | |
| Improved sanitation | 149 (6.4) |
| Unimproved sanitation | 2,192 (93.6) |
| Source of drinking water (n = 2,338) | |
| Improved | 1,121 (47.9) |
| Non-improved | 1,217 (52.1) |
| Disposal of child's stools when not using toilet (n = 2,417) | |
| Properly disposed | 635 (26.3) |
| Not properly disposed | 1,782 (73.7) |
| Birth weight (n = 2,267) | |
| LBW | 29 (1.3) |
| Normal | 257 (11.3) |
| Not weighted | 1,982 (87.4) |
| Age of child in months | |
| 0 | 314 (12.9) |

*(Continued)*

**Table 1.** (Continued)

| Variables | Frequency (%) |
|---|---|
| 1 | 438 (18.0) |
| 2 | 424 (17.4) |
| 3 | 433 (17.8) |
| 4 | 442 (18.2) |
| 5 | 382 (15.7) |
| Survey years | |
| 2011 | 1,248 (51.3) |
| 2016 | 1,185 (48.7) |

LBW, low birth weight

breastfeeding earlier than 6 months of age of children is associated with increased occurrence of diarrhea, fever, ARI, and poor nutritional outcomes like wasting and underweight.

Breastfeeding is a well-known child feeding option that provides immediate and long-term protection from infections through its rich content of immune factors, anti-microbial and anti-inflammatory agents, and nutrients. It can also prevent inoculation of disease-causing pathogens through contaminated liquids or foods and gastrointestinal damage from feeding liquids or solids foods that easy entry of infectious agents [4]. This finding revealed that children who had terminated EBF in between 0 and 3 or 4 and 6 months of age had increased chance of getting diarrhea episode, which is in-line with other studies which reported the protective effect of exclusive breastfeeding for the first six months from gastrointestinal infections [9–13, 24, 25]. Similarly, a study reported that a high breastfeeding performance index was associated with a low diarrhea risk [26]. This effect might be due to the fact that when babies used exclusive breastfeeding, they could be prevented from infection which resulted from contaminated feeding materials, water and foods. It has been seen that more than forty percent of diarrhea occurrence in children under 6 months of age can be prevented by implementing exclusive breastfeeding practices.

The study found that termination of EBF between 4 and 6 months of age were associated with increased symptoms of ARI. Similarly Dieterich et al indicated that infants ceased exclusive breastfeeding at 4 month were at greater risk of getting upper-respiratory tract infection than those children exclusively breastfed for 6 months [27]. Other studies also demonstrated similar effect of early EBF termination on respiratory illnesses [9, 25, 28, 29]. The effect of ceasing exclusive breastfeeding for the occurrence of ARI in children might be explained by the fact that ARI is the sign and symptom of many infections that could be resulted from contamination when the child exposed during initiation of complementary feeding. This study also revealed that breastfeeding according to the 2001 WHO recommendation of EBF for the first six months could reduce the burden of acute respiratory illnesses by 27%.

It is also found that termination of EBF earlier than 6 months were associated with adverse nutritional outcomes. Earlier termination of exclusive breastfeeding is linked with increased occurrence of childhood wasting or underweight. Nearly, 26% of childhood wasting and 23% underweight can be prevented if EBF was not terminated in the first six months of child life which is in agreement with a study done in Bangladesh [25]. On the other hand, Kramer and Kakuma noted that no adverse effects on babies growth have been documented with exclusive breastfeeding for six months in both developed and developing countries [11]. These facts also could be explained in that the nutritional status of babies are directly linked with their exposure status to infection. As described in the earlier paragraphs of this study, children who did

**Table 2. Breastfeeding, morbidity and nutritional status of under six month's children, 2011 and 2016 EDHS, Ethiopia.**

| Variables | Frequency (%) |
|---|---|
| Exclusive breastfeeding for the 1st 6 months (n = 2,433) | |
| EBF | 1,330 (54.7) |
| Not EBF | 1,103 (45.3) |
| EBF discontinuation time in months (n = 2,352) | |
| Not terminated up-to 6 | 1,330 (56.5) |
| 0–3 months | 507(21.6) |
| 4–6 months | 515 (21.9) |
| Diarrhea in the last 2 weeks (n = 2,428) | |
| Yes | 218(9.0) |
| No | 2,210 (91.0) |
| Fever in the last 2 weeks (n = 2,432) | |
| Yes | 359 (14.8) |
| No | 2,073 (85.2) |
| Acute respiratory illness (ARI) in the last 2 weeks (n = 2,428) | |
| Yes | 153 (6.3) |
| No | 2,275 (93.7) |
| Children with at least one morbidity (n = 2,427) | |
| Yes | 500 (20.6) |
| No | 1,927 (79.4) |
| Stunting (HAZ) (n = 2,108) | |
| Stunted | 272(12.9) |
| Normal | 1,836 (87.1) |
| Wasting (WHZ) (n = 2,080) | |
| Wasting | 295 (14.2) |
| Normal | 1,785 (85.8) |
| Underweight (WAZ) (n = 2,157) | |
| Underweight | 234 (10.8) |
| Normal | 1,923 (89.2) |
| Children with at least one adverse nutritional outcomes | |
| Yes | 598 (28.4) |
| No | 1,509 (71.6) |

EBF, exclusive breastfeeding; HAZ, height for age z-score; WHZ, weight for height z-score; WAZ, weight for age z-score

not exclusively breastfed have more risk of being infected with diarrhea and ARI than their counterparts. This intern has effect on the nutritional status of children who did not exclusively breastfed. However, in this study, childhood stunting and termination of EBF earlier than 6 months of age didn't show significant association which is in-line with other studies done in developing countries [25, 30, 31]. On the other hand, EBF is associated with decreased occurrence of stunting as revealed by some other studies [29, 32, 33]. Still the relation between stunting and EBF is appealing for further investigation. Through it is difficult to provide concrete justification for the lack of association between EBF and stunting, it might be due to the fact that stunting is an indicator of chronic undernutrition for which six months may not be sufficient time to see the exact effect of EBF in children under six months of age.

**Table 3. Effect of exclusive breastfeeding on adverse health outcomes among children, 2011 and 2016 EDHS, Ethiopia.**

| EBF cessation time | Adverse health outcomes | | | | | | | | | | | |
|---|---|---|---|---|---|---|---|---|---|---|---|---|
| | Diarrhea | | | Fever | | | ARI | | | At-least one adverse health outcomes | | |
| | COR [95% CI] | AOR [95% CI] | PAF % | COR [95% CI] | AOR [95% CI] | PAF % | COR [95% CI] | AOR [95% CI] | PAF % | COR [95% CI] | AOR [95% CI] | PAF % |
| Not terminated up-to 6 | 1 | 1 | | 1 | 1 | | 1 | 1 | | 1 | 1 | |
| Terminated between 0–3 months | 2.03 [1.13,3.62] | 1.95 [1.08,3.53]* | 42.5 | 1.43 [.92,2.22] | 1.56 [0.99,2.46] | 20.8 | 1.10 [0.51,2.32] | 1.12 [0.51,2.45] | 27.3 | 1.52 [1.05,2.21] | 1.66 [1.13,2.43]* | 26.0 |
| Terminated between 4–6 months | 3.81 [2.39,6.07] | 3.57 [2.19,5.83]** | | 1.70 [1.14,2.55] | 1.73 [1.11,2.68]* | | 2.67 [1.58,4.54] | 2.74 [1.61,4.65]** | | 2.15 [1.53,3.02] | 2.15 [1.48,3.11]** | |

CI, confidence interval; COR, crudes odds ratio; AOR, adjusted odds ratio; ARI, acute respiratory illness; PAF, population attributable fraction

* EBF discontinuation is significant at p-value less than 0.05

** EBF discontinuation is significant at p-value less than 0.001.

This study has both strengths and limitations. As a strength, we used two consecutive surveys data so as to get nationally representative sample. As a limitation, the DHS assess child feeding practice using a single 24 hours recall method. So relaying on this data to measure exclusive breastfeeding may not capture the true feeding practice of infants switching between EBF and mixed feeding at some points of time. According to WHO, children under five months of age can be classified as exclusively breastfed even when they have received early complementary feeding including traditional fluids [34]. Researchers also documented that the use of the 24 hours maternal recall method to measure EBF is less accurate measure than other methods such as the gold standard deuterium dilution dose-to-mother (DTM) method [35] and a single 24 hours recall method can also overestimate [36] or underestimate EBF.

## Conclusions

Exclusive breastfeeding cessation time had effect on the occurrence of childhood morbidity and adverse nutritional outcomes. The study concluded that termination of EBF earlier than six months of age was associated with increased occurrence of diarrhea, fever and ARIs. Earlier

**Table 4. Effect of exclusive breastfeeding on nutritional outcomes among children, 2011 and 2016 EDHS, Ethiopia.**

| EBF cessation time | Adverse health outcomes | | | | | | | | | | | |
|---|---|---|---|---|---|---|---|---|---|---|---|---|
| | Stunting | | | Wasting | | | Underweight | | | Had at-least one adverse nutritional outcomes | | |
| | COR [95% CI] | AOR [95% CI] | PAF % | COR [95% CI] | AOR [95% CI] | PAF % | COR [95% CI] | AOR [95% CI] | PAF % | COR [95% CI] | AOR [95% CI] | PAF % |
| Not terminated up-to 6 | 1 | - | | 1 | 1 | | 1 | 1 | | 1 | - | |
| Terminated between 0–3 months | 0.59 [0.31,1.14] | - | - | 2.18 [1.35,3.49] | 2.32 [1.45,3.74]** | 26.4 | 1.10 [0.61,1.99] | 1.11 [0.60,2.07] | 23.2 | 1.37 [0.94,1.99] | - | 9.2 |
| Terminated between 4–6 months | 0.88 [0.54,1.45] | - | | 1.33 [0.78,2.28] | 1.45 [0.84,2.49] | | 2.27 [1.37,3.76] | 2.30 [1.36,3.91]* | | 1.16 [0.79,1.69] | - | |

CI, confidence interval; COR, crudes odds ratio; AOR, adjusted odds ratio; ARI, acute respiratory illness; PAF, population attributable fraction

* EBF discontinuation is significant at p-value less than 0.05

** EBF discontinuation is significant at p-value less than 0.001.

termination of exclusive breastfeeding was also linked with increased occurrence of childhood wasting or underweight. The 2001 WHO recommendation, EBF for the first six months, could be the recalled child feeding practice in Ethiopia based on the evidence generated. It should be stressed on promotion of effective implementation of exclusive breastfeeding for the first six months of childhood life to reduce childhood morbidities and improve child growth performance.

## Supporting information

**S1 Table. STROBE 2007 (v4) Statement—Checklist of items that should be included in reports of cross-sectional studies.**
(PDF)

**S2 Table. Explanatory variables categorization and coding.**
(PDF)

## Acknowledgments

We thank ICF Macro-international for allowing us to use both the 2011 and 2016 Ethiopian Demographic and Health Survey data set for free.

## Author Contributions

**Conceptualization:** Dabere Nigatu, Muluken Azage.

**Formal analysis:** Dabere Nigatu, Muluken Azage, Achenef Motbainor.

**Writing – original draft:** Dabere Nigatu.

**Writing – review & editing:** Dabere Nigatu, Muluken Azage, Achenef Motbainor.

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
