## [Decision Letter · Decision Letter 0]

13 Aug 2019

PONE-D-19-14831

Effect of exclusive breastfeeding cessation time on childhood morbidity and adverse nutritional outcomes in Ethiopia: Analysis of the demographic and health surveys

PLOS ONE

Dear Dr Dabere Nigatu,

Thank you for submitting your manuscript to PLOS ONE. After careful consideration, we feel that it has merit but does not fully meet PLOS ONE’s publication criteria as it currently stands. Therefore, we invite you to submit a revised version of the manuscript that addresses the points raised during the review process.

ACADEMIC EDITOR: Be sure to include:

We would appreciate receiving your revised manuscript by September 30 th. To enhance the reproducibility of your results, we recommend that if applicable you deposit your laboratory protocols in protocols.io, where a protocol can be assigned its own identifier (DOI) such that it can be cited independently in the future. For instructions see: http://journals.plos.org/plosone/s/submission-guidelines#loc-laboratory-protocols

We look forward to receiving your revised manuscript.

Kind regards,

Massimo Ciccozzi

Academic Editor

PLOS ONE

Journal Requirements:

2. In ethics statement in the manuscript and in the online submission form, please provide additional information about the database used in your retrospective study. Specifically, please ensure that you have discussed whether all data were fully anonymized before you accessed them and/or whether the IRB or ethics committee waived the requirement for informed consent. If patients provided informed written consent to have their data used in research, please include this information.

Reviewers' comments:

Reviewer's Responses to Questions

**Comments to the Author**

1. Is the manuscript technically sound, and do the data support the conclusions?

Reviewer #1: Partly

Reviewer #2: Yes

2. Has the statistical analysis been performed appropriately and rigorously? 

Reviewer #1: I Don't Know

Reviewer #2: Yes

3. Have the authors made all data underlying the findings in their manuscript fully available?

Reviewer #1: Yes

Reviewer #2: No

4. Is the manuscript presented in an intelligible fashion and written in standard English?

Reviewer #1: Yes

Reviewer #2: Yes

5. Review Comments to the Author

Reviewer #1: INTRODUCTION:

- Page 5, line 77-78: support the sentence "Moreover, to the best of our knowledge, such studies are lacking in Ethiopia" with a reference.

METHOD:

section STUDY DESIGN AND SETTINGS:

- Page 6, line 91-92:

1) what is meant by enumeration area and population and housing census?

2) why were these two chosen as a cluster?

- Page 6, line 93-96:

1) If EA is a cluster and urban and rural areas are selected based on EA, then urban and rural are sub-clusters? Define all clusters better.

2) what is the connection between the first and second sampling phase? why have they been divided in this way?

3) “a fixed number of households per cluster were selected”, which clusters does it refer to? urban and rural area or AE and PHC, or all 4?

4)Which is the systematic random sampling technique?

section OUTCOME VARIABLES MEASUREMENT:

-Page 7, line 122-126: How was “less than-2 standard deviation” defined as a limit?

-Page 7, line 119,126,127:Did the children included in the study have both variables (at least one nutritional and at least one morbidity) or only the nutritional variable and only the morbidity variable?

RESULTS:

section SOCIO-DEMOGRAPHIC CHARACTERISTICS

-Page 9-11, Table 1: In the "Exposure variables measurement" section the following variables were considered which are not present in table 1: type of cooking fuel; sanitation facility; source of drinking water; disposal of child's stools when not using toilet. How were they considered in the study?

Reviewer #2: This paper is quite interesting and focuses its attention on a very important topic. Nevertheless, this study presents significant limitations, which are however well described and discussed in "Methods" and "Discussion".

- Table 1: data about household wealth status are presented. Further explanation in the text about this variable might be important.

- Table 3 and 4: for easier reading, I suggest to add p values to COR/AOR (95% CI) for each variable.

6. PLOS authors have the option to publish the peer review history of their article (what does this mean?). If published, this will include your full peer review and any attached files.

Reviewer #1: Yes: Cecilia De Flora

Reviewer #2: No

---

## [Author Response · Author response to Decision Letter 0]

16 Sep 2019

Authors’ response to reviewers’ comments: 

Response to reviewer #1 comments:

INTRODUCTION: 

Comment 1: - Page 5, line 77-78: support the sentence "Moreover, to the best of our knowledge, such studies are lacking in Ethiopia" with a reference.

Response: Thank you for the comment. The sentence stated in the referred lines was the authors’ argument about the lack of evidence on the current topic of interest (i.e. Effect of exclusive breastfeeding cessation time on childhood morbidity and adverse nutritional outcomes in Ethiopia).

METHOD: 

Section STUDY DESIGN AND SETTINGS:

Comment 1: - Page 6, line 91-92: 1) what is meant by enumeration area and population and housing census? 2) Why were these two chosen as a cluster?

Response 1: The 2007 population and housing census (PHC) is one of the censuses conducted in Ethiopia by the Central Statistical Agency of Ethiopia. It involves a complete enumeration of the populations of Ethiopia for basic information including age, sex, ethnicity, and residence. The 2007 PHC of Ethiopia created census frame used for enumeration. A total of 84,915 complete list of enumeration areas were created as a census frame. An enumeration area is a geographic area covering on average 181 households. The census frame contains information about the EA location, type of residence (urban or rural), and estimated number of residential households. Therefore, the enumeration areas created for 2007 census was used as sampling frame for the 2011 and 2016 Ethiopian Demographic and Health Surveys. In the surveys, an enumeration area was taken as a cluster. 

Hence, we added some points in the revised version of the manuscript to make it clear.

Response 2: Regarding the question “why were these two chosen as a cluster?” we want to clarify that it is the enumeration area that was taken as a cluster. The 2011 and 2016 Ethiopian Demographic and Health Survey are a national and sub-national representative surveys. To undertake these surveys, it is demanding to have a nationally representative sampling frame. Thus, the census frame generated for the 2007 population and housing census of Ethiopia is becoming the only readily available sampling frame to use for the surveys. 

We have added some points in the revised version of the manuscript.

Comment 2: - Page 6, line 93-96:

 1) If EA is a cluster and urban and rural areas are selected based on EA, then urban and rural are sub-clusters? Define all clusters better.

Response: Thank you for the comment. An enumeration area can be rural or urban but not both (rural and urban). There is no sub-clustering rather there is stratification of the EAs into rural or urban. Both rural EAs and urban EAs are selected independently. For example, for 2016 EDHS, 645 EAs (202 urban EAs and 443 rural EAs) were selected. This is done to ensure representativeness of indicators for both urban and rural population of Ethiopia. We have added some points in the revised version of the manuscript. 

Comment 3: - Page 6, line 93-96: 2) what is the connection between the first and second sampling phase? Why have they been divided in this way?

Response: In the first sampling phase, the EAs are selected from each stratum. Then, complete residential household listing was done for the selected clusters (i.e. EAs). In the second phase, after complete household listing, a fixed number of households per EAs were selected using systematic sampling procedure. Moreover, the details of EDHS sampling procedure is found and well stated in the original document which is available on The DHS Program web page, particularly Ethiopia DHS 2011 and 2016. (https://dhsprogram.com/Where-We-Work/Country-Main.cfm?ctry_id=65&c=Ethiopia&Country=Ethiopia&cn=&r=1)

Comment 4: - Page 6, line 93-96: 3) “a fixed number of households per cluster were selected”, which clusters does it refer to? Urban and rural area or AE and PHC, or all 4?

Response: Here the Authors want to refer the EAs since an EA is taken as a cluster.

Comment 5: - Page 6, line 93-96: 4)Which is the systematic random sampling technique?

Response: There is household listing in the selected EAs. Then, selection of a fixed number of households from list of households per EAs. At this stage, systematic sampling technique was applied to select households. 

Section OUTCOME VARIABLES MEASUREMENT:

Comment 1: -Page 7, line 122-126: How was “less than-2 standard deviation” defined as a limit?

Response: Thank you for the comment. The DHS data set had anthropometric z-scores generated through interpolation function. We defined less than minus two standard deviation as limit for undernutrition (stunting, wasting and underweight) based on global recommendations including WHO, UNICEF, and Food and Nutrition Technical Assistance (FANTA) recommendations

(https://www.fantaproject.org/sites/default/files/resources/FANTA-Anthropometry-Guide-May2018.pdf, https://www.who.int/childgrowth/standards/Technical_report.pdf?ua=1, https://www.who.int/nutrition/publications/anthropometry-data-quality-report/en/). Children with anthropometric z-scores below minus two standard deviation are considered as undernutrition namely stunting, wasting and underweight. 

Comment 2: -Page 7, line 119,126,127: Did the children included in the study have both variables (at least one nutritional and at least one morbidity) or only the nutritional variable and only the morbidity variable?

Response: Thank you for the comment. In this study, we considered both variables (children with at-least one morbidity and children with at-least one adverse nutritional outcome) independently for all children included in the study. 

RESULTS: 

Section SOCIO-DEMOGRAPHIC CHARACTERISTICS

 Comment 1::-Page 9-11, Table 1: In the "Exposure variables measurement" section the following variables were considered which are not present in table 1: type of cooking fuel; sanitation facility; source of drinking water; disposal of child's stools when not using toilet. How were they considered in the study?

Responses: Thank you for your concern. Previously, we considered these variables only in the regression analysis as potential confounders. But, now, in the revised version of the manuscript we have included them in table 1. Additionally, we provided the details of explanatory variables coding and categorization as supplementary files (S2 table)

Response to reviewer #2 comments:

General comment: This paper is quite interesting and focuses its attention on a very important topic. Nevertheless, this study presents significant limitations, which are however well described and discussed in "Methods" and "Discussion".

Comment 1: - Table 1: data about household wealth status are presented. Further explanation in the text about this variable might be important.

Response: Thank you for your comment. The comment accepted and we included in the revised version of the manuscript.

Comment 2: - Table 3 and 4: for easier reading, I suggest to add p values to COR/AOR (95% CI) for each variable. 

Response: Thank you for your comment. The comment accepted and we included in the revised version of the manuscript.

---

## [Editor Report · Decision Letter 1]

20 Sep 2019

Effect of exclusive breastfeeding cessation time on childhood morbidity and adverse nutritional outcomes in Ethiopia: Analysis of the demographic and health surveys

PONE-D-19-14831R1

Dear Dr.Dabere Nigatu ,

We are pleased to inform you that your manuscript has been judged scientifically suitable for publication and will be formally accepted for publication once it complies with all outstanding technical requirements.

With kind regards,

Massimo Ciccozzi

Academic Editor

PLOS ONE